# CIC-DUX4 Chromatin Profiling Reveals New Epigenetic Dependencies and Actionable Therapeutic Targets in CIC-Rearranged Sarcomas

**DOI:** 10.3390/cancers16020457

**Published:** 2024-01-21

**Authors:** Arnaud Bakaric, Luisa Cironi, Viviane Praz, Rajendran Sanalkumar, Liliane C. Broye, Kerria Favre-Bulle, Igor Letovanec, Antonia Digklia, Raffaele Renella, Ivan Stamenkovic, Christopher J. Ott, Takuro Nakamura, Cristina R. Antonescu, Miguel N. Rivera, Nicolò Riggi

**Affiliations:** 1Clinical Pathology Service, Department of Diagnostics, Geneva University Hospital, 1205 Geneva, Switzerland; 2Pediatric Hematology-Oncology Research Laboratory, Woman-Mother-Child Department, Lausanne University Hospital, University of Lausanne, 1011 Lausanne, Switzerland; 3Platform Genomics Technologies, Center for Integrative Genomics, Faculty of Biology and Medicine, Lausanne University Hospital, University of Lausanne, 1011 Lausanne, Switzerland; 4Experimental Pathology Service, Lausanne University Hospital, University of Lausanne, 1011 Lausanne, Switzerlandivan.stamenkovic@unil.ch (I.S.);; 5Department of Cell and Tissue Genomics, Genentech. Inc., South San Francisco, CA 94103, USA; 6Department of Histopathology, Central Institute, Valais Hospital, 1951 Sion, Switzerland; 7Department of Oncology, Lausanne University Hospital, University of Lausanne, 1011 Lausanne, Switzerland; 8Massachusetts General Hospital Cancer Center, Charlestown, MA 02129, USA; christopher.ott@mgh.harvard.edu (C.J.O.);; 9Department of Medicine, Harvard Medical School, Boston, MA 02115, USA; 10Broad Institute of MIT and Harvard, Cambridge, MA 02142, USA; 11Institute of Medical Science, Tokyo Medical University, Tokyo 160-0023, Japan; 12Department of Pathology, Memorial Sloan Kettering Cancer Center, New York, NY 10065, USA; antonesc@mskcc.org

**Keywords:** sarcoma, CIC-DUX4, p300, epigenetics, ChIP-seq

## Abstract

**Simple Summary:**

CIC-DUX4-rearranged sarcoma (CDS) is a rare and aggressive soft tissue tumor predominantly affecting young adults for which the only oncogenic driver is the CIC-DUX4 fusion protein, resulting from chromosomal rearrangements. The molecular mechanisms by which CIC-DUX4 drives CDS development remain largely unknown, hindering the development of targeted therapies. This study conducted genome-wide profiling of CIC-DUX4 DNA occupancy and associated chromatin states in CDS cell models and tumors. Our findings reveal CIC-DUX4 as a potent transcriptional activator at its binding sites, facilitated by direct interaction with the acetyltransferase p300. Significantly, inhibiting p300 impedes CDS tumor cell proliferation, suggesting a potential therapeutic avenue. This research enhances understanding of CIC-DUX4-mediated transcriptional regulation and proposes targeting p300 as a promising strategy for the clinical management of CDS.

**Abstract:**

CIC-DUX4-rearranged sarcoma (CDS) is a rare and aggressive soft tissue tumor that occurs most frequently in young adults. The key oncogenic driver of this disease is the expression of the CIC-DUX4 fusion protein as a result of chromosomal rearrangements. CIC-DUX4 displays chromatin binding properties, and is therefore believed to function as an aberrant transcription factor. However, the chromatin remodeling events induced by CIC-DUX4 are not well understood, limiting our ability to identify new mechanism-based therapeutic strategies for these patients. Here, we generated a genome-wide profile of CIC-DUX4 DNA occupancy and associated chromatin states in human CDS cell models and primary tumors. Combining chromatin profiling, proximity ligation assays, as well as genetic and pharmacological perturbations, we show that CIC-DUX4 operates as a potent transcriptional activator at its binding sites. This property is in contrast with the repressive function of the wild-type CIC protein, and is mainly mediated through the direct interaction of CIC-DUX4 with the acetyltransferase p300. In keeping with this, we show p300 to be essential for CDS tumor cell proliferation; additionally, we find its pharmacological inhibition to significantly impact tumor growth in vitro and in vivo. Taken together, our study elucidates the mechanisms underpinning CIC-DUX4-mediated transcriptional regulation.

## 1. Introduction

Chromosomal translocations often result in the formation of fusion proteins that can drive malignant transformation, leading to the emergence of various cancer types [1,2]. These aberrant proteins, also known as onco-fusion proteins, may acquire neo-morphic properties which are distinct from their parent counterparts, and frequently hijack the epigenetic machinery to induce cellular transformation from permissive precursor cells [3]. In keeping with this, the signaling and transcriptional regulatory mechanisms, as well as the oncogenic properties, controlled by the newly formed onco-fusion proteins, heavily rely on their fusion partners and originating cells [2].

Fusion proteins associated with transcription factors (TFs) are typically created by merging the DNA-binding domain of the TF with the transactivation domain from another protein. While they are likely to retain DNA-binding specificity, their genome-wide occupancy, co-factor preferences, and chromatin activity can differ from their corresponding wild-type (wt) counterparts, providing each fusion protein with distinct neo-morphic properties that vary across tumor entities [4]. This type of aberrant TF-mediated tumorigenesis is mainly observed in a subset of leukemias and solid tumors, particularly sarcomas [3,5] and relies on the expression of the fusion proteins in a permissive cellular context that creates a favorable environment for their oncogenic activity [6,7,8].

Several studies, including from our own group, have examined the biology and mechanisms involved in various translocated sarcoma models and described the distinctive features of specific onco-fusion proteins, including EWS-FLI1, EWS-ATF1, and EWS-WT1, which drive Ewing sarcoma, clear-cell sarcoma, and desmoplastic small round-cell tumors, respectively [4,8,9,10,11,12]. While different aberrant fusion proteins may share some functional properties, a detailed investigation of their activity remains critical to delineate their unique characteristics and their impact on the biology of each sarcoma model, including the less studied CDS tumors.

CIC-DUX4 sarcoma is an aggressive soft tissue tumor that prevalently develops in young adults, and is part of the CIC-rearranged sarcoma category defined by the 5th edition of the WHO classification of soft tissue and bone tumors [13]. Morphologically, CIC-DUX4 sarcoma is characterized by the proliferation of small blue round cells and defined as small blue round-cell tumors (SBRCT) [14,15]. In CDS, the gene *Capicua* (*CIC*) is fused to the double homeobox 4 gene (*DUX4*) to generate the onco-fusion gene *CIC-DUX4*. *CIC* is a ubiquitously expressed transcriptional repressor involved in inhibiting the MAPK pathway, whereas *DUX4* is a retrotransposed intronless gene located within the D4Z4 microsatellite repeated units that encodes for a pioneer TF expressed during early embryonic development and capable of activating gene transcription through p300 recruitment [16]. Given that *DUX4* is present as multiple duplications on either chromosome 4 or 10, the resulting fusion gene can be generated by two alternative genetic events between chromosome 19 (*CIC*) and either chromosome 4 or 10 (*DUX4*). The resulting CIC-DUX4 aberrant TF contains the majority of the N-terminal part of CIC, including its DNA binding domain, and a small portion of the C-terminal part of DUX4 which contain its transactivation domain [16], and is believed to represent the main genetic driver of this disease, given the lack of additional genetic alterations in the majority of CDS [17,18].

Interestingly, the functional conversion of CIC from a canonical repressor into a powerful transcriptional activator occurring in the context of CIC-DUX4 is sufficient to establish a CDS-specific tumorigenic program [17,18]. However, despite recent reports assessing CIC-DUX4 direct target genes [19] and CIC-DUX4 mechanisms of action [18,20], the precise molecular underpinnings allowing CIC-DUX4 to activate its direct target gene repertoire remains poorly understood, limiting the identification of new mechanism-based therapeutic options for this aggressive disease. Recent studies have started highlighting the potential of targeted therapies for these tumors [21], including the inhibition of the G2/M checkpoint regulator WEE1 [22], the combination of Trabectedin with the AKT/mTOR inhibitor NVPBEZ235 [23], and the chemical perturbation of the histone acetyltransferase p300 [20]. Although these works are paving the way for future therapeutic opportunities, the precise functional link between these pathways and CIC-DUX4 activity remains elusive.

In this study, we investigated the chromatin remodeling properties of CIC-DUX4 in CDS utilizing well-established cell line models and primary human tumors. Our analysis shows that CIC-DUX4 operates as a powerful transcriptional activator at its genomic binding sites, activating a gene expression program that characterizes primary CDS tumors. Furthermore, we combined genetic and pharmacological perturbations with proximity ligation assays to identify the direct interaction of CIC-DUX4 with p300 in vivo, and confirmed the critical role of this chromatin regulator in the establishment of the CDS oncogenic program. Finally, our findings provide compelling mechanistic evidence for the in vivo dependency of CDS tumors on p300 activity, and suggest a potential therapeutic intervention for this deadly malignancy.

## 2. Materials and Methods

### 2.1. CIC-DUX4 Sarcoma Primary Tumors

Three primary tumor specimens were collected with approval from the Institutional Review Boards (IRB) of Memorial Sloan Kettering Cancer Center (MSKCC, New York, NY, USA), protocol number 02-060. Informed written consent was obtained for all samples collected for this study. Samples were anonymized prior to analysis.

### 2.2. Cell Lines

The NCC-CDS1-X1(CDS1) and NCC-CDS2-C1(CDS2) cell lines were kindly provided by Tadashi Kondo (Division of rare cancer research head, National Cancer Center Research Institute, Tokyo, Japan). HEK293T cells were obtained from the ATCC. CDS1 and CDS2 cells were grown in RPMI medium (Gibco, Waltham, MA, USA) supplemented with 10% Fetal Bovine Serum (FBS) and 1% Penicillin-Streptomycin (PenStrep, Gibco) in adherent cell culture conditions. Cells were passed in RPMI medium (Gibco) supplemented with 20% KO serum (Gibco) one week prior experimentations. LentiX HEK293T, HeLa, and RDES cells were grown in Dulbecco’s Modified Eagle Medium (DMEM) supplemented with 10% FBS and 1% PenStrep. All cells were cultured at 37 °C with 5% CO_2_. Cells were maintained and split every 2–4 days (at approximately 80% confluence). Cells were cryopreserved in cryogenic medium [90% FBS with 10% dimethyl sulfoxide (DMSO)] into liquid nitrogen for long-term storage. RD-ES Ewing’s sarcoma cell line was obtained from ATCC, cells were cultured in RPMI medium supplemented with 15% FBS and 1% PenStrep and sub-cultured twice per week.

### 2.3. Lentiviral Generation

Lentivirus was produced in LentiX HEK293T (Clontech-Takara Bio Europe, Saint-Germain-en-Laye, France)) by FuGene6 (Promega, Madison, WI, USA) transfection with gene delivery vector and packaging vectors GAG/POL and VSV plasmids. Growth medium was changed after 6–8 h from the initial transfection. Viral supernatants were collected 72 h after transfection and concentrated using LentiX concentrator (Clontech-Takara Bio Europe, Saint-Germain-en-Laye, France), following the manufacturer’s instructions. Virus containing pellets were resuspended in PBS and added dropwise on cells in presence of media supplemented with 6 μg/mL polybrene. CDS1 and CDS2 lentivirally infected cells were selected using G418 at a final concentration of 250 μg/mL during 96 h.

### 2.4. Knock-Down of CIC-DUX4 and p300 by shRNA Transfection

shRNA constructs for P300 knockdown were purchased from Genecopoeia (Genecopoeia, Rockville, MD, USA) (Cat#HSH088760- LVRU6GP, sense sequence: CCAATGGTGGTGATATTAATC and GGATTATGACTTGTGTATCAC, antisense sequence: GATTAATATCACCACCATTGG and GTGATACACAAGTCATAATCC). Negative control shRNA was purchased from Genecopoeia. SiRNAs for depletion of CIC-DUX4 were purchased from Qiagen (Qiagen, Hilden, Germany) (Hs_DUX4_11, sequence: CAGGCGCAACCTCTCCTAGAA) or from Sigma-Aldrich (Sigma-Aldrich, Saint-Louis, MO, USA) (SASI_Hs02_0036094, sense sequence: CCGGCGCCCUGGUCUGCAC [dT][dT], antisense sequence: GUGCAGACCAGGGCGCCGG [dT][dT]). Negative control siRNA (AllStars) was purchased from Qiagen. Oligonucleotides were transfected with INTERFERin siRNA transfection reagent (Polyplus, Illkirch, France) in 150 mm plates using siRNA duplexes at a final concentration of 12 nM and a reverse transfection protocol according to manufacturer recommendations. Cells were harvested 3 days after transfection.

### 2.5. RNA Extraction, cDNA Synthesis, and Real-Time Quantitative PCR

RNA was extracted using the RNeasy RNA isolation kit (Qiagen, cat#74104, Hilden, Germany), DNAse treatment was performed following the manufacturer’s instructions. For cDNA synthesis, 500 ng of template total RNA were reverse transcribed using the high-capacity cDNA reverse transcription kit (Thermofisher scientific, Waltham, MA, USA, cat#4368814). Real-time qPCR amplification was performed using PowerUp SYBR^®^ Green Master Mix (Applied Biosystems, Waltham, MA, USA, cat#A25742) and specific PCR primers in a QuantStudio 5 Real-Time PCR System (ThermoFisher Scientific, Waltham, MA, USA). Oligonucleotides used are provided in Appendix A. Relative quantification of each target, normalized to an endogenous control (GAPDH or TBP), was performed using the comparative Ct method (Applied Biosystems, Waltham, MA, USA). Error bars indicate SD of three technical replicates.

### 2.6. Western Blot Analysis

Cell lysis, SDS-PAGE, and immunoblotting were performed using standard procedures. A measure of 40 μg of proteins were loaded per lane for Western blot and samples underwent electrophoresis through an 8% polyacrylamide gel at 125 mV for 1 h 30. Primary antibodies used for Western blotting are listed in Appendix A. Secondary antibodies were HRP-conjugated goat anti-mouse (GE healthcare, Chicago, Il, USA) and goat anti-rabbit (Dako-Agilent, Santa Clara, CA, USA) antibodies. Protein signals were revealed by SuperSignal West Pico and Femto Chemiluminescent Substrate (Thermo Scientific, Waltham, MA, USA) and captured with the Fusion FX device (Vilber Lourmat, Marne-la-Vallée, France).

### 2.7. Proximity Ligation Assay (PLA)

The proximity ligation assay was performed using a Duolink II Fluorescence PLA kit (Olink Bioscience, Uppsala, Sweden), as instructed by the manufacturer. Cells were seeded at 70% confluence in 0.2 cm2 dishes, fixed in 4% paraformaldehyde in PIPES buffer for 13 minutes at RT, and permeabilized with 0.3% triton in PBS for 3 minutes. Primary antibodies, anti P300 (cell signaling, Danvers, MA, USA), and anti DUX4 P4H2 (Thermofisher scientific were used with a 1:2000 dilution. PLA amplification was labeled with Alexa Fluor 594 (Olink Bioscience, Uppsala, Sweden). Slides were counterstained with DAPI, mounted, and imaged using the Zeiss (Oberkochen, Germany) Confocal Fluorescent Microscope LSM710, with oil immersion objective 63X, NA 1.4. For each channel the pin hole was set to 0.9 AU. For each sample, the Z-stack was acquired, with a line averaging 2 passages and with a sampling in the XYZ according to the optimal Nyquist criteria. Before analysis the Z-stack was converted with maximum intensity projection. The resulting images were analyzed using ImageJ software version 1.52 R 26 October 2019 (http://rsbweb.nih.gov/ij/, accessed on 31 October 2019) with a script that defines as region of interest (ROI) the DAPI stained nuclei and counts the included PLA fluorescent foci. For statistical analysis fluorescent foci were counted for each sample in 4 different fields each containing an average of 8–10 cells.

### 2.8. Chromatin Immunoprecipitation Followed by Sequencing (ChIP-seq)

ChIP assays were carried out on approximately 2–5 million cells or 5–10 mg frozen tumor tissue per sample and per epitope, following the protocol described previously [8]. Briefly, chromatin from formaldehyde-fixed cells was fragmented to a size range of 200–600 bases using a Branson 250 sonicator. Solubilized chromatin was immunoprecipitated with the indicated antibodies (listed in Appendix A) overnight at 4 °C. Antibody–chromatin complexes were pulled down with protein G-Dynabeads (Invitrogen, Waltham, MA, USA), washed, and then eluted. After crosslink reversal, RNase A, and proteinase K treatment, immunoprecipitated DNA was extracted with AMP Pure beads (Beckman Coulter, Brea, CA, USA). ChIP DNA was quantified with Qubit (Life technologies, Carlsbad, CA, USA). A measure of 1 to 5 ng of ChIP DNA was used to prepare sequencing libraries, and ChIP DNA and input controls were sequenced with the Hi-seq Illumina (San Diego, CA, USA) genome analyzer.

### 2.9. TF and Histone Mark ChIP-seq Data Analysis

Single-end reads were trimmed using TrimGalore version (v) 0.6.4 (https://github.com/FelixKrueger/TrimGalore, accessed on 31 October 2019) and aligned to the human genome assembly GRCh37 (hg19) using STAR v 2.5.0a58. Multiple matches, as well as regions present in the ENCODE project DAC Exclusion List Regions (dataset ENCSR636HFF), were removed and a maximum of three redundant reads were kept for subsequent analyses.

Significantly enriched genomic regions for TF and histone mark ChIP-seq were detected using MACS2 v 2.2.659 with an FDR set to 0.05. To reduce the amount of false-positive enrichment, we applied another method to identify significantly enriched genomic regions60. Briefly, the whole genome was split into 400-bp bins and the scores (log2 of the bins reads counts) of these regions were calculated for both the IP and the input samples. A pseudo count was added to avoid NAs. Bins were then split into quantiles based on the mean scores of the IP and input ((IP + Input)/2); for each quantile, the Z-scores were calculated using the mean and SD of the bins fold change (log2(IP)-log2(Input)) within the quantile. *p*-values were calculated using the pnorm function in R and adjusted with the Benjamini–Hochberg method. In each quantile, bins with an FDR below 0.001 were considered as enriched. Only the MACS2 peaks that intersected with the enriched genomic regions identified by the above-mentioned method were kept for further analysis. The BEDtools v 2.27.1 intersect command, with the -u option of unicity, was used to extract the enriched regions common between the two methods61. This process was applied to each ChIP-seq sample separately. For the generation of the DUX4 reference peak set, all DUX4 peaks identified in the four ChIP-seq experiments (two replicates each of the CDS1 and CDS2 cell lines) were merged, as well as regions overlapping within +/− 150 bp. A total of 810 DUX4 peaks were identified. The peaks within +/− 1 kb of a known gene’s TSS (ucsc_RefFlat_07_08_ 2016) were labeled as TSS. The remnant peaks were evaluated for H3K4me3 enrichment as above and labeled as TSS if H3K4me3-enriched regions were identified in five ChIP-seq experiments, giving in total 428 TSS-associated DUX4 peaks. All the other peaks were labeled as Distal (*n* = 382). For the genomic distributions, the Distal peaks were annotated as intragenic or intergenic according to the ucsc_RefFlat_07_08_2016 list. IGV was used to visualize ChIP-seq tracks62.

CPM-normalized BigWig files were generated using bamCoverage63, with an extension of ChIP-seq reads to 300 bp fragment length, and used for the ChIP signal heat maps and composite plots (generated with the deepTools scripts computeMatrix, plotHeatmap, and plotProfile). Heat maps were sorted based on the DUX4 mean signal in the four cell line replicate ChIP-seq experiments.

For ChIP-seq peak score calculation, the read counts per peak was calculated using bedTools with the –c option, normalizing to 20 M reads per sample (the approximate mean reads across all samples) and scaling to 500 bp peak width (the approximate DUX4 reference peak mean width). Each peak score is given by log2(IP + 1)-log2(Input + 1), where +1 is an additional pseudo-count, mean values of the replicates. For each dataset, the mean peak read count in the input sample was used as the lower threshold. The peaks in the IP sample with a read count below the threshold were set to zero. Correlations between ChIP-seq scores were calculated by Pearson’s correlation coefficient (r).

Centered, 500 bp extended peak regions were searched for De Novo motif enrichment using HOMER v 4.11.164, applying the 500 bp target sequence length and chopify options. The De Novo motifs were then clustered using the «compareMotifs.pl» script from HOMER. To find motif half-site enrichment, a second round of motif analysis was performed using shorter motif length (4, 5, 6 nt).

All figures, except heat maps and composite plots, were generated using R (R: A language and environment for statistical computing. R Foundation for Statistical Computing, Vienna, Austria, http://www.R-project.org/, v 4.1.0, accessed on 30 June 2020)).

### 2.10. RNA-seq Differential Expression Analysis

For RNA-seq analysis, total RNA was extracted from frozen tumor tissue using the TRIzol reagent (Invitrogen) or from cells using the RNeasy Mini kit (Qiagen, Hilden, Germany) according to the manufacturer’s instructions. Minimum 100 ng used for preparing sequencing libraries using the TruSeq mRNA stranded kit (Illumina, San Diego, CA, USA) and sequencing was performed on a Hi-Seq Illumina Genome Analyzer (100 bp single-read). Samples used for RNA-seq profiling included siRNAQ-, siRNAS-, or siCTRL-treated CDS1 cell lines and DMSO- or dCBP-1 100 nM-treated CDS2 cell lines.

Single-end reads were trimmed using TrimGalore v 0.6.4 and aligned to the human genome and transcriptome assembly GRCh37 (hg19) using STAR v 2.5.0a. Transcript quantification was performed using rsem v 1.3.068. The resulting counts matrix was used for subsequent analysis. Analysis was performed in R; gene filtering was performed based on the rule of 1 count per M (cpm) in at least 1 sample; library size scaling was performed using TMM normalization (EdgeR package v 3.32.1)69, Robinson Bioinformatics (2010)); log-transformation was performed using the Limma voom function70. Normalized data were batch-corrected using the Limma removeBatchEffect function and unwanted variation was removed with RUVr (RUVseq package v 1.24.0)71 whenever needed. Statistically significant differentially expressed genes were identified from log-transformed, TMM-scaled values using lmFit of the Limma package by fitting a linear model on the selected samples. P-values were adjusted for multiple testing using the Benjamini–Hochberg correction. Batch and/or RUVr-computed factors were added in the design matrix.

Analysis was performed by considering both cell lines (CDS1 and CDS2) and both siRNAs (siRNAQ and siRNAS) as replicates. Those four samples were tested against CDS1 and CDS2 DMSO-treated samples. All RNAseq-related figures were generated using R v 4.1.0 (http://www.R-project.org/, R Foundation for Statistical Computing, Vienna, Austria, accessed on 30 June 2020)).

### 2.11. Cell Viability Assays

CDS1 and CDS2 cells viability tests using CellTiter-Glo^®^ luminescent cell viability assay (Cat# G7571, Promega Corporation, Madison, WI, USA) were performed by plating 20,000 cells/well in triplicates in 96-well cell culture plates. Cell viability was measured after 3, 5, and 7 days. Endpoint luminescence was measured on a SpectraMax M5 plate reader (Molecular Devices, San José, CA, USA).

### 2.12. In Vitro p300 Pharmacological Inhibition

For pharmacological targeting of p300, the A-485 chemical inhibitor [24] was dissolved in DMSO. A total of 5000 CDS1 cells and 15,000 CDS2 cells were plated in each well as quadruplicates in a 96-wells’plate and allowed to adhere for 24 h in KO serum complemented medium. The A-485 drug was added at a final concentration of 0.01, 0.05, 0.1, 0.25, 0.5, 1, and 2 μM for CDS1, and 5 and 10 μM were added for CDS2 treatment. The according volumes of DMSO were added to control wells and cells were treated with 2 μM Staurosporine in 4 supplementary wells. The fraction of viable cells was determined using the CellTiter-Glo^®^ luminescent cell viability assay (Cat# G7571, Promega Corporation, Madison, WI, USA), as described by the manufacturer. To generate dose–response curves, data were normalized by setting the average value of Staurosporine and DMSO control wells to 0% and 100% viability, respectively. The replicate values for each dilution point were averaged and the EC_50_ values for each compound were generated in Prism GraphPad by fitting normalized data to a sigmoidal curve model of linear regression. D5 timepoint EC_50_ values of each cell line were reported in the published cell screening study, from which the same protocol was followed [24].

### 2.13. In Vitro p300 Degrader Assay

dCBP-1 P300 degrader was obtained from (MedChemExpress, Monmouth Junction, NJ, USA https://doi.org/10.1016/j.chembiol.2020.12.004) [25]. Cells were cultured for 1–3 days in RPMI containing 20% KnockOut™ Serum Replacement, 1% penistrep, then stimulated with 50 or 100 nM dCBP-1 or vehicle (DMSO) and harvested after 24, 48, 72, and 96 h for Western blot analysis and total RNA extraction. For viability test cells were seeded in 60 mm plates at a concentration of 200,000/plate and cultured for 3 days in RPMI containing 20% KnockOut™ Serum Replacement, 1% penistrep. After controlling the number of cells/well, stimulation with 50 or 100 nM dCBP-1 or vehicle (DMSO) was started and cell count was performed in triplicates at regular intervals over a period of time of 4 days. Trypan blue exclusion was used to evaluate cell viability. Cell counts were performed using an automated cell counter Countess II (Thermo-Fisher, Waltham, MA, USA).

### 2.14. In Vivo Studies

All mice experiments were approved by the Cantonal Veterinary Office, authorization number VD3021 and VD3437. NOD/SCID c Gamma (NSG) mice aged 5–7 weeks were purchased from Jackson Laboratory, Bar Harbor, ME, USA (stock number 005557). All mice were monitored three times a week for tumor development, and sacrificed using CO_2_ inhalation and cervical dislocation, when the tumor reached 1000–1500 mm^3^ volume or at the end of the experiment. Every time, an accurate necropsy of the mice was performed to assess local tumor growth and metastatic spread.

### 2.15. In Vivo A-485 Treatment

Here, 2 × 10^6^ CDS1 cells were resuspended in 100 uL of 1X PBS prior to bilateral subcutaneous injection in the subcutaneous suprascapular region of NSG mice. The experiment included a cohort of eleven mice, which were allocated between the control-treated (*n* = 6) or the drug-treated (*n* = 5) group to normalize their size before starting the treatment. A-485 (MedChemExpress, Cat# HY-107455, Monmouth Junction, NJ, USA) was prepared daily and administered intraperitoneally at a dose of 50 mg/kg twice daily. When the first tumors started to be visible, mice were treated with either 50 mg/kg of A-485 or solvent for 15 days. All mice were monitored daily for signs of distress and were weighed three times a week. Tumor size was measured three times a week with a caliper, and tumor volume was calculated according to the following equation: (length × width^2^)/2 = tumor volume [mm^3^]. Control-treated mice received the solvent used for oral administration. Mice were sacrificed, as described above, the day after the last dose of A-485.

### 2.16. Quantification and Statistical Analysis

Statistical analysis of wet lab experiments (Student’s *t* test, 2-way ANOVA, Mann–Whitney) were performed through Prism GraphPad Software 8.00. All statistical tests and sample numbers are disclosed in respective Figure Legends.

## 3. Results

### 3.1. CIC-DUX4 Preferentially Binds to TGAAT- and GGAA-Containing Motifs and Associates with an Active Chromatin Signature in CDS

To understand how CIC-DUX4-mediated epigenetic changes drive CDS tumor development, we first sought to identify CIC-DUX4 DNA binding sites and their associated genome-wide chromatin states in different CDS models. To this end, we profiled CIC-DUX4 chromatin occupancy and a panel of histone modifications in two well-established human CDS cell lines (CDS1 and CDS2). Both cell lines are derived from CDS patient samples and express the CIC-DUX4 fusion protein [26,27]. (Appendix A). To profile CIC-DUX4 binding sites, we performed chromatin immunoprecipitation (ChIP) sequencing in CDS1 and CDS2 cells using an antibody targeting the C-terminal part of wt DUX4 (given that both CDS cell lines do not express a significant level of endogenous DUX4 protein (Appendix A). For each cell line, two independent biological replicates were generated and combined to create a robust list of 810 overlapping CIC-DUX4 binding sites shared between the two lines (Figure 1A), which were subsequently used for all further analyses (Appendix A). Notably, we found that 53% of CIC-DUX4 peaks were located at gene promoters (TSS), while the rest were distributed throughout intra- and inter-genic regions (Figure 1A). This is in stark contrast to the genomic distribution of other sarcoma-driving fusion proteins (including EWS-FLI1 [8], EWS-ATF1 [4], and EWS-WT1 [28]), which preferentially bind to distal sites, but is reminiscent of the genomic distribution of wt CIC occupancy in normal cells [29,30].

Next, to determine the chromatin states associated with CIC-DUX4 occupancy we profiled both CDS cell lines and three primary CDS tumors for the major histone modifications, H3K4me1, H3K27ac, and H3K4me3. Two biological replicates were generated for each histone mark and cell line (CDS1 and CDS2), whereas one replicate was generated for each of the 3 primary CDS samples (CIC1, CIC2, and CIC3). Both distal and TSS regions bound by CIC-DUX4 were found to be associated with active histone marks (Figure 1B). Distal CIC-DUX4 peaks were enriched for the active enhancer marks H3K4me1 and H3K27ac, and CIC-DUX4 bound TSSs were similarly associated with the active promoter marks H3K4me3 and H3K27ac (Figure 1B). Notably, the three primary tumors CIC1, CIC2, and CIC3 displayed similar active chromatin signatures at the corresponding CIC-DUX4 bound regions (Figure 1B,D), confirming the transcriptional activator role of the fusion protein in CDS. This observation is in strong contrast to the reported repressive function of wt CIC, which is known to recruit histone deacetylases and inhibit transcription [29].

Given that CIC-DUX4 retains the DNA binding domain of wt CIC, we next asked whether the fusion protein also conserves the corresponding DNA binding motif, or if the association with DUX4 provides the fusion protein with distinct binding properties. To this end, we surveyed distal CIC-DUX4 sites (*n* = 382) for underlying DNA motifs and found enrichment for the Hbp1, bHLH, and HMBOX1-related de novo motifs (Figure 1C right). Both Hbp1 and bHLH motifs contain the canonical TGAAT sequence recognized by the wt CIC, suggesting that CIC-DUX4 retains the binding preferences of wt CIC. Interestingly, the known motif analysis revealed enrichment for ETS TFs motifs, including ETV4, ETV2, and EWS-FLI1 (Figure 1C left). CIC-DUX4 has been shown to directly induce ETV1, ETV4, and ETV5 expression [31]; one potential explanation for this result is the co-binding of a subset of genomic sites by the fusion protein and its own targets.

Altogether, these results suggest that the functional conversion from a transcriptional repressor to an activator displayed by the fusion protein is not related to a major shift in its DNA binding properties or genomic distribution [29,31], and may include the co-binding of a fraction of sites with TFs from the ETS family [30].

### 3.2. CIC-DUX4 Chromatin Occupancy Leads to H3k27acetylation and Target Gene Activation

To further characterize the chromatin remodeling functions of CIC-DUX4, we next employed a siRNA-mediated knock-down strategy to deplete the fusion protein in the CDS1 cell line. Two independent siRNAs targeting the 3′ end of DUX4 (DUX4 siRNA-1 and DUX4 siRNA-2) were utilized. This approach resulted in a notable reduction in CIC-DUX4 protein (over 90%) and mRNA levels (30–50%), as compared to cells transfected with non-targeting control siRNAs (Figure 2A). In line with the transcriptional activator role of CIC-DUX4, the expression levels of a selected panel of known CIC-DUX4 target genes showed a significant decrease following depletion of the fusion protein (Figure 2B). A similar trend was also observed when repeating the same experiment in the CDS2 cell model using siRNA-2 (Appendix A).

To further investigate how CIC-DUX4 depletion influences the chromatin states associated with its binding sites, we conducted ChIP-seq on CDS1 cells transfected separately with both DUX4-targeting siRNAs hairpins. Our analysis revealed a reduction in H3K27ac signal across the majority of CIC-DUX4-bound sites, which was more pronounced at distal regulatory elements (Figure 2C–E). This reduction in H3K27ac signal at CIC-DUX4-bound regions supports the active role played by the fusion protein in establishing and maintaining active chromatin and transcriptional states in CDS tumors.

To further assess the impact of CIC-DUX4 depletion on global gene expression, we conducted RNA-seq analysis on CIC-DUX4-depleted CDS1 cells and CDS2 cells using two different siRNAs, for a total of four replicates knock-down samples, that we compared to one sample per cell-line treated with DMSO. Using an adjusted *p*-value cutoff set to 0.05 and a log2 fold-change cutoff set to 1 we identified 697 and 850 genes exhibiting significant decreases and increases upon CIC-DUX4 depletion, respectively (Figure 2F). Since the marked number of genes showing increased expression following CIC-DUX4 knock down may seem in contrast with the activator role of the fusion protein, we then looked for their association with CIC-DUX4 ChIP-seq peaks. This analysis revealed that the vast majority of direct CIC-DUX4 target genes (83/93) displayed decreased expression upon the fusion protein depletion, including a set of well-established CDS-tumor specific markers (ETV1, ETV5, DUSP4, ETV4, SPRED3) [29] (Figure 2F andAppendix A). Interestingly, functional annotation of differentially expressed genes using Gene Ontology (GO) analysis identified a clear difference in their biological function (Figure 2G). Whereas genes showing decreases (697) were involved in cell proliferation and cell cycle control, transcripts induced upon CIC-DUX4 depletion (850) were associated with cell migration and differentiation pathways. The lack of direct target genes among the induced transcripts suggests that the majority of these genes are the result of the establishment of a new gene expression program rather than direct repression by CIC-DUX4, as observed for other sarcoma-associated fusion proteins [8].

Collectively, our analyses support the notion that CIC-DUX4 operates as a potent chromatin and transcriptional activator, directly contributing to the establishment of an oncogenic program favoring cell proliferation on migration and differentiation.

### 3.3. CIC-DUX4 Directly Interacts with the p300 Acetyltransferase at Its Binding Sites

Given the major changes in H3K27ac signals observed in CIC-DUX4-depleted cells, and the reported interaction between the C-terminus of wt DUX4 and the chromatin remodeler p300, we next focused on the possible interaction between these two proteins in the context of CDS tumor cells. To explore this hypothesis, we assessed the physical interaction between the CIC-DUX4 and p300 proteins within CDS cell nuclei using a proximity ligation assay (PLA). PLA experiments were conducted using anti-p300 and anti-DUX4 antibodies to stain CDS1 nuclei; these revealed a significant number of interaction foci. This number was significantly higher than in the control experiment, where the same cells were assessed using IgG and anti-p300 antibodies (median number of foci 1.273 for IgG vs. 10.57 for p300, *p* value < 0.0001; Figure 3A,B).

To further substantiate our findings at the chromatin level, we conducted ChIP-seq profiling for p300 in CDS1 cells and assessed whether p300 and CIC-DUX4 chromatin peaks co-occur in the genome of CDS cells. In accordance with our PLA results, we identified a significant enrichment of p300 ChIP-seq signals at chromatin regions occupied by CIC-DUX4 (Figure 3C,D). In summary, our findings provide support to the notion that, while retaining the DNA binding properties of wt CIC, CIC-DUX4 operates as a transcriptional activator, at least in part through its direct interaction with p300.

### 3.4. Genetic and Pharmacological Repression of p300 Activity Reduces the In Vitro and In Vivo Growth Ability of CDS Cells

As CIC-DUX4 may recruit p300 to its binding sites in order to induce the expression of its target genes repertoire, we sought to assess whether targeting p300 may represent a viable therapeutic approach to decrease CDS cells proliferation and tumorigenicity. To this end, we specifically decreased the P300 mRNA levels in CDS cells using a shRNA-mediated knockdown strategy, and measured the ensuing changes in cellular proliferation. Lentiviral-mediated knockdown using two independent shRNAs (P300 shRNA-1 and shRNA-2) in CDS1 cells achieved a significant reduction in p300 protein (60–95%) mRNA levels (60–70%), as compared to control cells infected with a non-targeting shRNA (Figure 4A,B).

We therefore quantified the changes in proliferation rate and/or viability of p300-depleted CDS1 cells at day 3, 5, and 7 post-infection. While control shRNA-infected cells maintained a proliferative rate similar to their untreated counterparts, p300-depleted cells showed a gradual reduction in their proliferation ability, which reached 70–75% at day 7, confirming the marked dependency of these tumor cells on p300 activity (Figure 4C). To substantiate our findings, and validate the specificity of our shRNA strategy, we repeated these experiments using the well characterized p300 pharmacological inhibitor A-485. Recently, p300 inhibition using this agent has been reported to decrease CDS tumor cell proliferation and target gene expression [20]. Similar to the results obtained using our shRNA approach, we also observed a significant reduction in CDS1 and CDS2 tumor cells proliferation upon 5 days of A-485 treatment (57% with 500 nM and 73% with 1 uM concentration, Figure 4D). Similar results were observed in CDS2 cells with A-485 treatment for 5 days (Appendix A). To better characterize the degree of sensitivity displayed by CDS cells toward p300 inhibition, we took advantage of a study reporting the effect of A-485 on 124 different cancer cell lines treated with this compound for 3, 4, and 5 days, and where the sensitivity threshold was set at 1μM or below [24]. We tested our CDS cell lines using the same experimental conditions, and integrated our results on CDS1 and CDS2 cells into their screening data. By drawing a dose–response curve and normalizing our results to a positive control represented by the same cells treated with the chemotherapeutic agent staurosporine, we calculated an absolute effective concentration of 50 (EC_50_) for our two cell lines (Figure 4E). CDS1 cells showed the highest sensitivity toward A-485, with an EC_50_ of 330 nM after five days of treatment, whereas CDS2 cells displayed an EC_50_ of 891 nM (Figure 4E). As compared to the reported 37 cell lines treated for 5 days in the previous study, our CDS cell lines scored among the top 15 most sensitive cancer lines, ranking 4 and 11 for CDS1 and CDS2 models, respectively (Figure 4F). Furthermore, we assessed the impact of inhibiting p300 on the expression of CIC-DUX4 target genes. Consistent with the changes in cell proliferation, the expression levels of a panel of established CIC-DUX4 target genes also showed a marked reduction at 5 days of A-485 treatment (Figure 4H). Interestingly, similar to p300 knockdown, CIC-DUX4 protein levels also decreased upon A-485 treatment in CDS1cells (Figure 4G), suggesting that p300 could potentially regulate CIC-DUX4 protein stability by its acetyltransferase activity.

Given the potential translation impact of our findings, we decided to perform an orthogonal and innovative approach to explore the role of p300 in CDS cells by using a highly potent and selective p300 protein degrader, dCBP-1 [25]. Both CDS1 and CDS2 lines were treated with two different concentrations of dCBP-1 (50 and 100 nM), which induced a significant reduction in p300 protein levels at 24 h of treatment (Figure 5A). Similar to our observation with the p300-targeting shRNAs and A-485 compound, dCBP-1 also led to a reduction in the fusion protein levels, associated with a concomitant decrease in the mRNA levels of a panel of CIC-DUX4 target genes, which closely recapitulated the transcriptional changes observed upon CIC-DUX4 depletion (Figure 5B). Furthermore, we investigated whether the CIC-DUX4 target genes demonstrated any similarity in response to dCBP-1 treatment. Interestingly, RNA-seq analysis of dCBP-1-treated (100 nM) CDS2 cells confirmed that the behavior of CIC-DUX4 target genes resembled that observed during CIC-DUX4 knockdown (Figure 5C and Appendix A), validating the p300-mediated CIC-DUX4 activity in CDS tumors. Additionally, dCBP-1 treatment altered the growth and viability of CDS1 and CDS2 cells, as compared to DMSO-treated control cells (Figure 5D). Interestingly, treatment of the Ewing’s sarcoma cell line RDES with similar concentrations of dCBP-1 also yielded a considerable decrease in p300 protein levels, but no significant difference is tumor cell proliferation and/or viability was observed (Appendix A), suggesting the specific effect of p300 inhibition on CDS tumors growth.

Given that the genetic and pharmacological perturbation of p300 activity both led to a marked reduction in tumor cell proliferation in vitro, we next sought to determine whether p300 inhibition could represent a viable strategy to decrease CDS tumor growth in vivo. To this end, 2 million CDS1 cells were injected bilaterally into the suprascapular region of NOD-SCID gamma KO mice. At 4 weeks after injection, when small tumors developed at all injection sites, mice were randomized to account for tumor size differences, and 50 mg/kg of A-485, or the corresponding volume of the carrier, were administrated intraperitoneally twice per day to the test (*n* = 5 mice) and control (*n* = 6 mice) groups, respectively. Mice were monitored three times per week, and tumor volume and weight were scored after 14 days of treatment, when A-485-treated mice showed a reduction of 70% in tumor size and 56% in tumor volume (0.09 g and 0.1 cm^3^ reduction, respectively), as compared to the control group (Figure 5E). Importantly, while A-485 treatment had a marked effect on tumor growth, we did not observe a significant change in the animal’s weight during the course of the experiment (Figure 5F). 

Our results support the notion of an exquisite dependency of CDS tumor cells toward p300 inhibition. Given the high overlap in transcriptional changes observed between CIC-DUX4 depletion and p300 pharmacological degradation, it stands to reason that the anti-tumorigenic effect observed upon p300 depletion may be linked to its functional cooperation with CIC-DUX4 to control a panel of shared target genes involved in tumor growth.

## 4. Discussion

Sarcoma-associated fusion proteins drive the development of distinct tumor types arising in a variety of anatomical locations, and involving different epigenetic factors. Yet these chimeric proteins also share a number of functional features that underlie their ability to transform permissive precursors cells. Among these, the most notable is the acquisition of neo-morphic properties altering their chromatin activity, and enabling the establishment of tumor specific programs. In keeping with this, TFs such as FLI1 and ATF1 acquire pioneering properties and bind otherwise-inaccessible genomic regions when fused to EWS in Ewing and clear cell sarcoma, respectively [4,8]. Similarly, SS18-SSX expression in synovial sarcoma alters the composition and chromatin binding profile of the BAF complex, inducing an oncogenic transcriptional signature that supports tumor growth [32]. Our study shows an analogous mechanism in CDS, where the fusion of the DNA binding domain of CIC to the transactivation domain of DUX4 converts a transcriptional repressor into a powerful activator, concomitantly turning a well-established tumor suppressor into a potent oncogene.

To better understand the functional consequences of this event, in this study we leveraged two human patient-derived cell lines and three primary tumors (Appendix A) to investigate the chromatin remodeling patterns associated with CIC-DUX4 expression. Using a combination of chromatin profiling and functional perturbations studies, we defined a robust set of direct binding sites (shared between cell lines and showing similar chromatin patterns in primary tumors), confirmed the activator role of CIC-DUX4 at these sites, and established the corresponding target gene repertoire. Interestingly, motif analyses of the DNA sequences underlying a fraction of these genomic regions revealed enrichment for ETS TFs motifs. Given that CIC-DUX4 has been shown to directly induce the expression of ETV1, ETV4, and ETV5 [31], which all belong to the ETS TF family, a plausible scenario could be that the fusion protein operates along a spatiotemporal trajectory, starting with the activation of direct targets controlled by canonical HLH motifs, and followed by the induction of additional genes regulated by the co-binding of CIC-DUX4 with its own targets, including ETV1, 4, and 5.

Despite playing critical roles across all stages of tumor evolution, aberrant TFs are hardly targetable using standard pharmacological strategies. Current approaches trying to bypass this limitation include modulation of their downstream target genes, or targeting catalytically active binding partners of the fusion proteins. Since DUX4 has been shown to recruit p300 through its C-terminal portion, which is retained in CIC-DUX4, we explored the possibility that the fusion protein may directly hijack p300 at its binding sites to induce chromatin activation, and evaluated p300 inhibition as a viable therapeutic strategy for the clinical management of these tumors. In keeping with this, we report the direct interaction between CIC-DUX4 and p300 in CDS tumor cells, and confirm the role of this chromatin regulator in sustaining the oncogenic transcriptional program induced by the fusion protein. Our chromatin profiles show that p300 and CIC-DUX4 co-localize at a large fraction of the fusion protein binding sites in the CDS1 and CDS2 lines, a result further substantiated using PLA as an orthogonal approach. Consistent with this notion, inhibition of p300 activity in CDS cells by genetic and pharmacological approaches, recapitulated the transcriptional changes observed upon CIC-DUX4 depletion, and resulted in tumor growth decreases in vitro and in vivo. Altogether these results add mechanistic insight to the observation reported by Bosnakovski, D. et al. concerning the reduction in CDS tumor growth following p300 inhibition [20], and provide a new tangible therapeutic option that consists of a selective and potent p300 degrader [25].

An intriguing result of our study is the decrease in CIC-DUX4 protein levels observed upon the genetic or pharmacological inhibition of p300 activity. Although a general reduction in transcriptional expression is to be expected under similar conditions, the lack of corresponding decreases in *CIC-DUX4* transcript levels suggests that alternative mechanisms may underlie these observations. One potential hypothesis warranting future investigation is that direct acetylation of CIC-DUX4 by p300 increases the stability of the fusion protein, as observed for other oncogenic TFs [33,34]. In this scenario, inhibiting p300 activity could represent a synergistic strategy that would reduce the transactivation ability of CIC-DUX4, while decreasing the global levels of the fusion protein itself.

Our current work also presents some limitations inherent to biology and to these tumors. First, given the rarity of available CIC-DUX4 sarcoma samples and in vitro models in our study, we used two well-established cell lines (out of the four distinct cell lines reported in the literature), and three primary CIC-DUX4 sarcoma frozen samples. Thus, further validation of our findings in additional lines is warranted before their translation into clinical studies. A second limitation of our current work is the lack of assessment of the long-term in vivo effects of p300 inhibitors/degraders on mice survival and pharmacological toxicity, which will be highly informative before the clinical translation of these findings. Finally, future work should also focus on detailing the interaction between the endogenous CIC-DUX4 and wt CIC proteins in our cell lines, since the current anti-CIC antibodies do not discriminate between these two proteins, which share the majority of the CIC amino-acid sequence. Indeed, the precise role of wt CIC in the context of CDS remains to be investigated, particularly in regard to the potential mechanisms allowing CIC-DUX4 to override the repressor function of CIC. Given that CIC and CIC-DUX4 share the same DNA-binding motif, it stands to reason that the two TFs may compete with each other for the same binding regions and target genes within CDS tumor cells. Future experimental models in which the endogenous wt CIC gene is tagged using a CRSPR editing approach will be ideal in untangling the interaction between these two proteins in the context of CDS cells. Understanding if and how wt CIC and CIC-DUX4 functionally interact in the context of CDS will represent a major step forward in deciphering the pathogenesis of these tumors and beyond, given that CIC inactivating mutations are known to fuel the progression and dissemination of a large panel of cancer types [35].

## 5. Conclusions

In summary, using human primary CIC-DUX4 tumors and derived cell lines, we generated the genome-wide profile of CIC-DUX4 DNA occupancy and associated chromatin activity states in CDS. Our results reveal how fusing the C-terminal portion of DUX4 to the N-terminus of CIC converted a well-established tumor suppressor into a powerful oncogene. From a functional standpoint, this event turns a transcriptional repressor into an activator, which directly controls an oncogenic gene expression program. Combining genetic and pharmacological perturbations with proximity ligation assays, we also demonstrate the interaction of CIC-DUX4 with p300 in vivo, and confirm the critical role of this chromatin regulator in supporting the chromatin remodeling properties of CIC-DUX4. Finally, our findings provide compelling mechanistic evidence for the in vivo dependency of CDS tumors on p300 activity, and suggest a potential therapeutic intervention for this deadly malignancy.

## Figures and Tables

**Figure 1 cancers-16-00457-f001:**
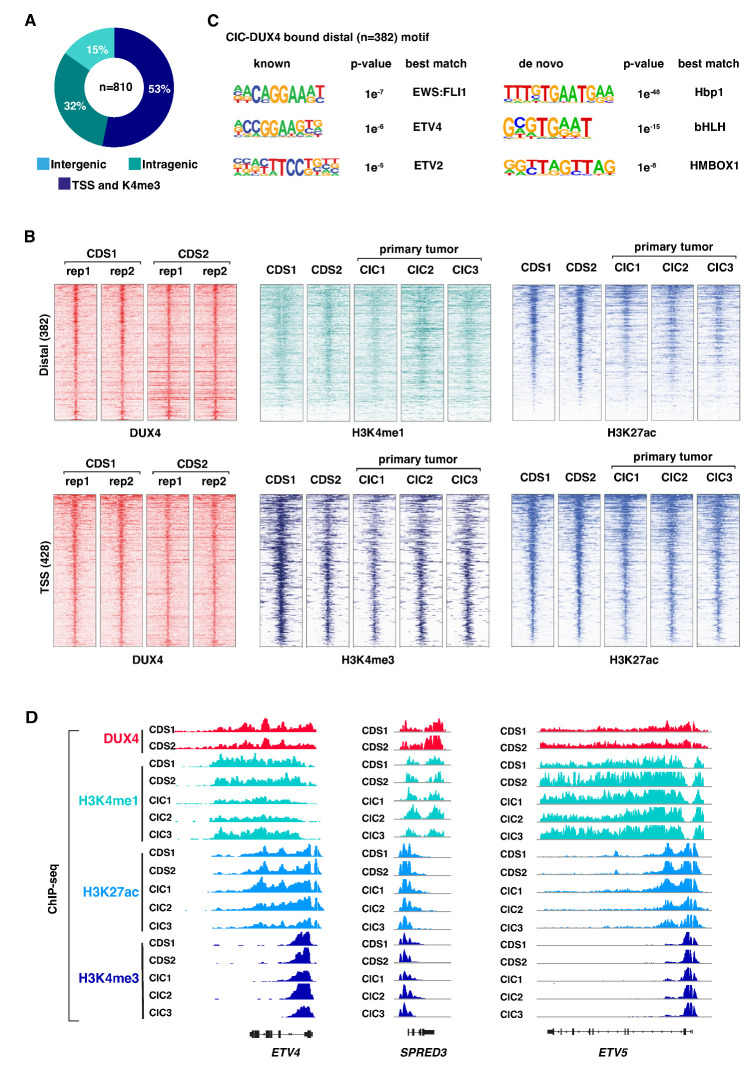
CIC-DUX4 associates with active chromatin attributes: (**A**) Pie chart illustrating the distribution of CIC-DUX4 binding sites in two CDS cell lines, CDS1 and CDS2. (**B**) Heatmaps depicting ChIP-seq signal densities for DUX4, H3K4me1, H3K4me3, and H3K27ac in CDS1 and CDS2 cell lines, along with three CDS primary tumors (CIC1-3). The top panel represents distal sites, while the bottom panel represents transcription start sites (TSSs). Each panel displays 10-kilobase windows centered on identified CIC-DUX4 binding sites from Figure 1A (**C**) Analysis of de novo and known DNA motif enrichment at distal CIC-DUX4 binding sites in CDS cell lines, as presented in Figure 1A. (**D**) Representative examples of CIC-DUX4 target gene loci, enriched for H3K4me1, H3K4me3, and H3K27ac ChIP-seq signals in CDS1 and CDS2 cell lines, as well as in three CDS primary tumors (CIC1-3).

**Figure 2 cancers-16-00457-f002:**
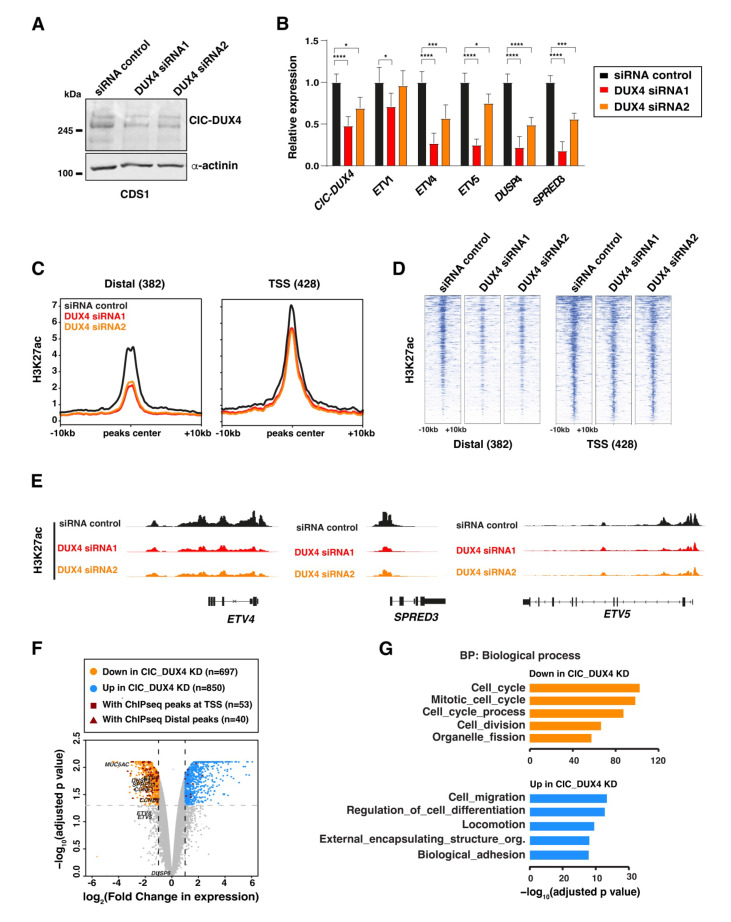
CIC-DUX4 depletion abrogates the H3K27ac signal and alters target gene expression: (**A**) Western blot depicts the depletion of the CIC-DUX4 protein in CDS1 cell line upon its transfection with two different siRNA constructs (siRNA1 and siRNA2). (**B**) RT-qPCR analysis of the expression levels for a panel of established CIC-DUX4 target genes with fusion protein depletion in CDS1 cells. Data are presented as mean +/− SD, with *n* = 3 per group (single asterisk *p* < 0.05, triple asterisk *p* < 0.001, quadruple asterisk *p* < 0.0001 using two-way ANOVA). (**C**) Composite plot shows CIC-DUX4 ChIP-seq signals in the CDS1 cells transfected with either DUX4-targeting or non-targeting control siRNAs. A 10 kb window centered on CIC-DUX4 binding sites. (**D**) Heatmaps depicting the changes in H3K27ac signals at the CIC-DUX4 binding sites in CDS1 cells transfected with either DUX4-targeting or non-targeting control siRNAs. (**E**) Representative ChIP-seq tracks showing chromatin changes at select loci upon CIC-DUX4 knockdown in the CDS1 cell line. (**F**) Volcano plot depicting gene expression changes in both CDS1 and 2 cell lines upon CIC-DUX4 knock-down. (**G**) Functional gene ontology categories (GO: Biological Process) for the differentially regulated genes upon CIC-DUX4 depletion. The uncropped bolts are shown in Appendix A.

**Figure 3 cancers-16-00457-f003:**
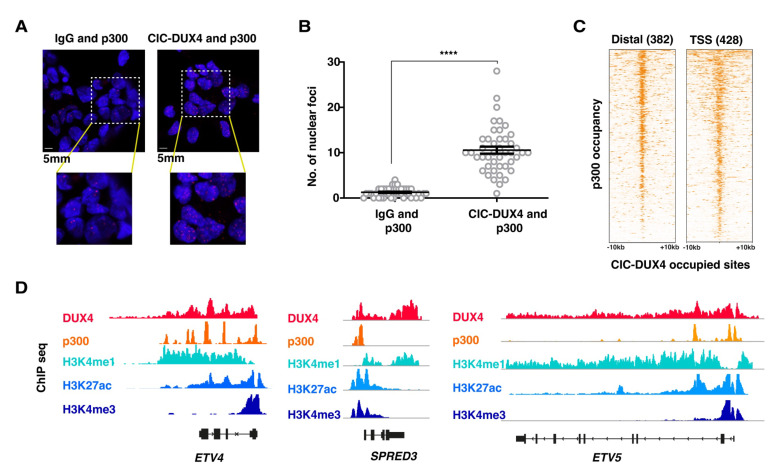
CIC-DUX4 and p300 proteins interact with each other and share chromatin occupancy: (**A**) Representative micrographs of PLA signals in the CDS1 cell line, assessed using either DUX4- targeting antibodies (right panels), or isotype-matched IgG (left panels). (**B**) Dot plot quantification of the number of interaction foci detected in the CDS1 cell lines. Data are presented as mean +/− SD. Statistical analysis was performed using student *t*-test (quadruple asterisk *p* < 0.0001). (**C**) Heatmaps showing the presence of p300 ChIP-seq signals at CIC-DUX4 binding sites in the CDS1 cell line. A 10 kb window centered on both distal (*n* = 382) and TSS (*n* = 428) CIC-DUX4 binding sites. (**D**) Representative ChIP-seq tracks showing chromatin CIC-DUX4 and p300 chromatin co-occupancy as well as H3K4me1, H3K27ac, and H3K4me3 signals at select loci in the CDS1 cell line.

**Figure 4 cancers-16-00457-f004:**
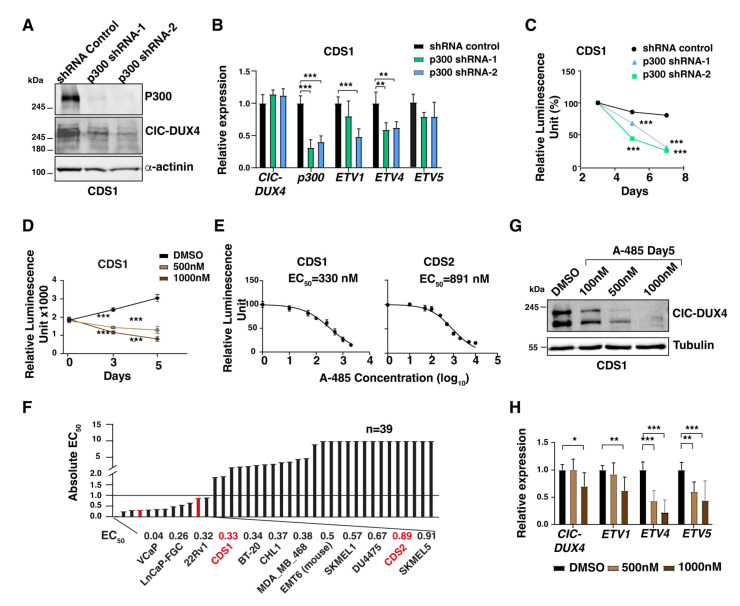
CDS tumor cells are sensitive to p300 depletion and pharmacological inhibition: (**A**) Western blot depicts the depletion of the p300 protein in CDS1 cell lines upon their transfection with two different siRNA constructs (shRNA-1 and shRNA-2). (**B**) RT-qPCR analysis for CIC-DUX4, EP300, and a panel of fusion protein target gene expression in CDS1 cell line transfected with two different p300-targeting siRNA constructs, or control siRNA. Data are presented as mean +/− SD, with *n* = 3 per group (double asterisk *p* < 0.01, triple asterisk *p* < 0.001 using two-way ANOVA). (**C**) Plot depicting the decrease in CDS1 cell proliferation upon p300 depletion, as assessed through relative luminescence. Data are presented as the means of relative luminescence unit with *n* = 4 per group (triple asterisk *p* < 0.001 using two-way ANOVA). (**D**) Changes in CDS1 proliferation upon treatment with increasing concentration of A-485, as assessed through luminescence. Data are presented as mean +/− SD, with *n* = 4 per group (triple asterisk *p* < 0.001 using two-way ANOVA). (**E**) Dose–response curve of CDS1 CDS2 after 5 days of treatment with A-485. (**F**): Bar plots show the EC_50_ values toward p300 inhibition by A-485 for a panel of cell lines, including CDS1 and CDS2 (red). (**G**) Western blot showing the decrease in CIC-DUX4 protein levels in CDS1 cells treated with increasing concentrations of A-485. (**H**) RT-qPCR analysis for CIC-DUX4 and a panel of fusion protein target genes in the CDS1 cell line treated with two different concentrations of A-485. Data are presented as mean +/− SD, with *n* = 3 per group (single asterisk *p* < 0.05, double asterisk *p* < 0.01, triple asterisk *p* < 0.001 using two-way ANOVA). The uncropped bolts are shown in Appendix A.

**Figure 5 cancers-16-00457-f005:**
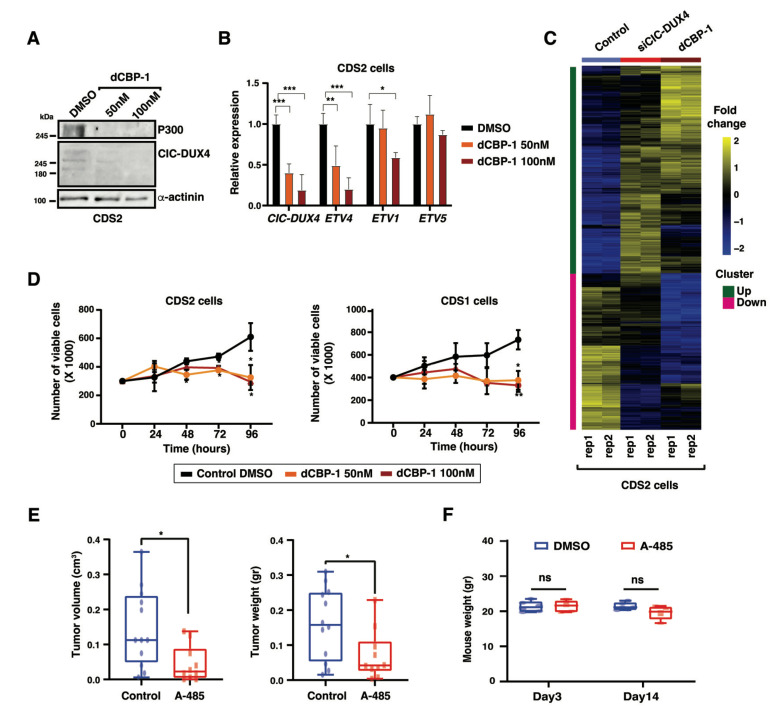
Pharmacological inhibition of p300 abrogates in vivo tumorigenicity: (**A**) Western blot depicts the decrease in p300 protein levels in CDS2 cells treated with increasing concentrations of the p300 protein degrader dCBP-1. (**B**) RT-qPCR analysis of the expression levels for CIC-DUX4 and a panel of fusion protein target genes in the CDS2 cell line treated with two different concentrations of dCBP-1. Data are presented as mean +/− SD, with *n* = 3 per group (single asterisk *p* < 0.05, double asterisk *p* < 0.01, triple asterisk *p* < 0.001 using two-way ANOVA). (**C**) Gene expression z-scores heatmaps in CDS2 cells for the transcripts showing upregulation and downregulation in CDS2 cell line upon CIC-DUX4 depletion and dGP1 treatment. (**D**) Plots depicting the decrease in CDS1 (right panel) and CDS2 (left panel) cell proliferation upon treatment with two different concentrations of dCBP-1, as assessed through relative luminescence. Data are presented as mean +/− SD, with *n* = 4 per group (single asterisk *p* < 0.05 using two-way ANOVA). (**E**) Box plots depict the changes in tumor volume (left panel) and tumor weight (right panel) observed for CDS1 mouse xenografts upon in vivo treatment with the p300 inhibitor A-485, as compared with mice treated with equivalent volume of the corresponding solvent. Data are presented as mean +/− SD. A-485 (*n* = 10) and control (*n* = 12). The difference between tumor volumes and weights were analyzed by Mann–Whitney U test (single asterisk *p* < 0.05). (**F**) Box plots depict the variation of mice weight for A-485 treated and control groups after 3 days and 14 days of treatment. Statistical analyses were performed using two-way ANOVA (ns *p* > 0.05). Data are presented as mean +/− SD. A-485 (*n* = 5) and control (*n* = 6). The uncropped bolts are shown in Appendix A.

## Data Availability

The data presented in this study are openly available in Gene Expression Omnibus (GEO) database under the series accession number GSE248117 (https://www.ncbi.nlm.nih.gov/geo/query/acc.cgi?acc=GSE248117, accessed on 17 November 2023 and the token # cfulsqmgxtwlpan) which contains the datasets GSE248040 ChIP-seq) and GSE248116 (RNA-seq). The public human genome sequence data (GRCh37, hg19) used in this study is available through the Ensembl genome browser (https://grch37.ensembl.org/Homo_sapiens/Info/Index, accessed on 17 November 2023). Raw and processed sequencing files for all cell line-based experiments have been deposited in GEO under super series number GSE248117. For primary tissue data, processed data only are available from GEO. The raw sequenced data have been deposited on ZENODO (https://zenodo.org/records/10141898, accessed on 17 November 2023). Primary raw data are not public but can be made available upon request.

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
