# Peer review of "CIC-DUX4 Chromatin Profiling Reveals New Epigenetic Dependencies and Actionable Therapeutic Targets in CIC-Rearranged Sarcomas"

_cancers, 2024, doi:10.3390/cancers16020457_

Round 1

Reviewer 1 Report

Comments and Suggestions for Authors

First, congratulations on this exceptional work. It is exciting to see advancements in mechanistic biology that point to clinically actionable targets in a rare disease.  Bravo!! This manuscript was a pleasure to read. It was very well-written with appropriate content in the intro to succinctly summarize complex biology and set the stage for the experiments described. The paper told a story that was easy to follow, compelling, and thought-provoking; I am already interested to hear how p300 inhibition (or degradation) turns out clinically.  I have a few very minor suggestions for consideration:

-If you have data showing target inhibition from your in vivo work, it would be nice to include, along with CIC-DUX4 fusion expression, to recapitulate what you see in vitro.

-line 538, double check the number of mice included. Figure 5E and legend shows n=10 and n=12, but text says n=5.

-line 480, there is an extra "in"

-In discussion, you could consider mentioning ongoing clinical trial investigating p300 inhibitor (NCT05488548; this is a dual BET and p300 inhibitor) or whether the CDK9 inhibitor approach could be effective in this disease context; but this could also be explored in future work!

Author Response

Dear Reviewer 1,

please find attached to this text the PDF named "Response_reviewer 1" where you will find the point-by point response to your comments.

Thank you very much for the time you dedicated to our manuscript. 

Arnaud Bakaric 

Reviewer 2 Report

Comments and Suggestions for Authors

The paper reports a genome-wide profiling of two CIC-DUX4 translocated sarcomas. Overall the data are sound and in line with literature data.

Nevertheless, some issues need to be addressed prior publication:

The authors state that an intriguing result of their paper is that p300 inhibition affects CIC-DUX4 protein levels but does not affect CIC-DUX4 transcript expression. This result is not shown and RT-qPCR using CIC-DUX4 specific primers must be provided.

To corroborate the notion that CIC-DUX4 occupancy is reminiscent of that of wild type CIC, the authors should provide ChIP-Seq and occupancy data also for wild type CIC.

To support the sentence “RNA-seq analysis of dCBP1-treated (100 nM) CDS2 cells confirmed that the behavior of CUC-DUC4 target genes resembled that observed during CUC-DUX4 knockdown (Fig 5C), validating the p300-mediated CUC-DUX4 activity in CDS tumors (Results, line 520-522) the authors should provide the intersection (VENN diagram) of the genes differentially expressed in CIC-DUX4 knockdown cells and dCBP1-treated cells.

Some statements should be downtoned. For instance, the sentence “In summary, our findings provide support to the notion that, while retaining the DNA binding properties of wt CIC, CIC-DUX4 operates as a transcriptional activator through its direct interaction with p300” (Results, lines 463-465) is an overstatement as their results are just correlative and suggest that p300 is likely to play a role in conferring CIC-DUX4 transactivating functions. Similarly, in discussion the authors state “Our study identifies an analogous mechanism in CDS, where the fusion of the DNA binding domain of CIC to the transactivation domain of DUX4 converts a transcriptional repressor into a powerful activator, concomitantly turning a well-established tumor suppressor into a potent oncogene”. Their paper does not really “identifies” the reported mechanism but just corroborates a well-established concept. The sentence should be downtoned.

The authors should provide detailed figure legends and double check figure labels (e.g. Figure 4 G and H are inverted).

Author Response

Dear Reviewer 2,

please find attached to this text the PDF named "response_reviewer 2" where you will find the point-by point response to your comments.

Thank you very much for the time you dedicated to our manuscript. 

Arnaud Bakaric 

Reviewer 3 Report

Comments and Suggestions for Authors

The authors  Bakaric and colleagues investigate the role of CIC-DUX4 and associated chromatin states in CIC-rearranged sarcomas in cell models and tumors. In particular they observed that CIC-DUX4 represents a potent transcriptional activator and its binding site interact directly with acetyltransferase p300. In this regard, inhibiting P300 could represent a promising way to impairing CIC-rearranged sarcomas tumor growth.

The manuscript is interesting and well organized. The findings could represent a starting point for further researches aimed at improving the management of CIC-rearranged sarcomas.

The manuscript would benefit from the followings:

1.       Latest WHO should be referenced: WHO Classification of Tumours. In Soft Tissue and Bone, 5th ed.; IARC Press: Lyon, France, 2020; Volume 3, p. 368, ISBN 978-92-832-4502-5.

2.       A graphical abstract underling the mechanism of action of CIC-DUX4, acetyltransferase p300 ant its possible impairing should be included.

3.       Clinicopathological information about the 3 parimary tumor specimens used for the analysis should be included

4.       Study limitations should be reported

Minor revisions are requested

Author Response

Dear Reviewer 3,

please find attached to this text the PDF named "response_reviewer 3" where you will find the point-by point response to your comments.

Thank you very much for the time you dedicated to our manuscript. 

Arnaud Bakaric 

Reviewer 4 Report

Comments and Suggestions for Authors

This well-designed research study generates in vitro and in vivo data from CIC-DUX4 fusion tumors and derived cell lines. The aim of the study is to better understand the transcriptional regulation mediated by the oncogene CIC-DUX4 and its interaction with the chromatin regulator p300 (histone acetyltransferase).

The results of the study demonstrate the dependence of CIC-DUX4 rearrangement sarcomas on p300 activity and suggest a therapeutic role in targeting p300.

Minor Comments/questions:

1.     Sarcomas with CIC-DUX4 rearrangement mostly affect young adults. Do you expect older or very old patients with CIC-DUX4 rearranged sarcoma to behave biologically the same way?

2.     It is amazing how quickly the powerful oncogenic program of CIC-DUX4 fusion could be inhibited by p300 blockade. Would you expect this to happen in clinical practice as well?

Author Response

Dear Reviewer 4,

please find attached to this text the PDF named "Response_reviewer 4" where you will find the point-by point response to your comments.

Thank you very much for the time you dedicated to our manuscript. 

Arnaud Bakaric 

Reviewer 5 Report

Comments and Suggestions for Authors

While this is a very interesting topic, and there is a critical need for better understanding of the disease biology of CIC-DUX4, and identification of therapeutic targets, the results of this paper are based on only two cell lines of CIC-DUX4, there are barely any reasonable controls in any of the experiments which seriously impacts the scientific soundness of the results. The discussion and abstract are overstating the impact of this work. p300 might be an interesting target, but there is still substantial pre-clinical work to be done before moving forward with this.

Author Response

Dear Reviewer 5,

please find attached to this text the PDF named "Response_reviewer 5" where you will find the point-by point response to your comments.

Thank you very much for the time you dedicated to our manuscript. 

Arnaud Bakaric 

Round 2

Reviewer 2 Report

Comments and Suggestions for Authors

The authors have addressed most of the concerns.

I believe that, although odd in the format, the figure showing the intersection between the down-regulated genes in CIC-DUX4-depleted and dCBP1-treated CDS cells should be included in the manuscript as a supplementary figure.

As for inversion Figures 4G and 4H, their description is inverted in the text (lines 505-501, see below).

"Consistent with the changes in cell proliferation, the expression levels of a panel of established CIC-DUX4 target genes also showed a marked reduction at 5 days of A-485 treatment (Figure 4G). Interestingly, similar to p300 knockdown, CIC-DUX4 protein levels also dereased upon A-485 treatment in CDS1cells (Figure 4H), suggesting that p300 could potentially regulate CIC-DUX4 protein stability by its acetyltransferase activity".

Author Response

Dear reviewer 2,

Thank you for your comments. Please find attached to this message a PDF we our responses. 

Best regards,

Arnaud Bakaric 

Reviewer 5 Report

Comments and Suggestions for Authors

The conclusion is still an overstatement. Mice should not be sacrificed before being followed for survival. Tumor size and mice weight are week surrogate markers (very weak) for survival and toxicity.

Author Response

Dear reviewer 5,

thank you for your last comments on our work. Please find attached to this message a PDF file with our response. 

Best regards,

Arnaud Bakaric 

Round 3

Reviewer 5 Report

Comments and Suggestions for Authors

ok to accept